# Promotion of Plant Growth in Arid Zones by Selected *Trichoderma* spp. Strains with Adaptation Plasticity to Alkaline pH

**DOI:** 10.3390/biology11081206

**Published:** 2022-08-12

**Authors:** Juan Pablo Cabral-Miramontes, Vianey Olmedo-Monfil, María Lara-Banda, Efrén Ricardo Zúñiga-Romo, Elva Teresa Aréchiga-Carvajal

**Affiliations:** 1Unidad de Manipulación Genética, Laboratorio de Micología y Fitopatología, Facultad de Ciencias Biologicas, Universidad Autónoma de Nuevo León, San Nicolás de los Garza 66451, Nuevo León, Mexico; 2Departamento de Biología, Division de Ciencias Naturales y Exactas, Universidad de Guanajuato, Guanajuato 38116, Gto., Mexico; 3FIME-Centro de Investigación e Innovación en Ingeniería Aeronáutica (CIIIA), Universidad Autónoma de Nuevo Leon, Av. Universidad s/n, Ciudad Universitaria, San Nicolás de los Garza 66455, Mexico

**Keywords:** arid soil, plant promotion, pH regulation, *Sorghum bicolor*, desert, alkalinity tolerance

## Abstract

**Simple Summary:**

Productivity in agriculture is affected by nutrient-deficit soil, a phenomenon caused by anthropogenic activities or by geographical location, such as in arid zones. The pH affects the availability of chemical compounds, which is why it is an elementary parameter for the survival of organisms. These conditions on the soil cause the inhibition of microorganism growth. *Trichoderma* are filamentous fungi located in a great diversity of ecosystems. They are commonly associated with plants and carry out fundamental activities in the development, biological control, and nutrition of plants. The study of areas with extreme conditions represents a valuable source of new strains with adaptations to environmental demands; locating sites where there are fungi with tolerance, evolution events, and adaptations to biotic and abiotic conditions provides evidence that there are microorganisms with the potential to improve soils and generate a positive effect on vegetable crops.

**Abstract:**

*Trichoderma* species are filamentous fungi that support plant health and confer improved growth, disease resistance, and abiotic stress tolerance. The objective of this study is to describe the physiological characteristics of the abundance and structure of *Trichoderma* model strains from arid zones and evaluate and describe their possible adaptation and modulation in alkaline pH. The presence of biotic factors such as phytopathogens forces farmers to take more actions such as using pesticides. In addition, factors such as the lack of water worldwide lead to losses in agricultural production. Therefore, the search for biocontrol microorganisms that support drought opens the door to the search for variations in the molecular mechanisms involved in these phenomena. In our case, we isolated 11 tested *Trichoderma* fungal strains from samples collected both from the rhizosphere and roots from two endemic plants. We probed their molecular markers to obtain their identity and assessed their resistance to alkaline conditions, as well as their response to mycoparasitism, plant growth promotion, and drought stress. The findings were worthy of being analyzed in depth. Three fungal taxa/species were grouped by phylogenetic/phenotypic characteristics; three *T. harzianum* strains showed outstanding capabilities to adapt to alkalinity stress. They also showed antagonistic activity against three phytopathogenic fungi. Additionally, we provided evidence of significant growth promotion in *Sorghum bicolor* seedlings under endemic agriculture conditions and a reduction in drought damage with *Trichoderma* infection. Finally, beneficial fungi adapted to specific ambient niches use various molecular mechanisms to survive and modulate their metabolism.

## 1. Introduction

Anthropogenic activities cause soil-limiting conditions; places are susceptible to becoming arid and semi-arid zones characterized by low or no bioavailable organic matter and limited nutrient content [1]. In these places, drought is the result of the adverse effects of global warming, and the onset of these factors determines the low yield of agricultural production [2]. In these areas, live plants that adapt to adverse biotic and abiotic conditions assisted by microorganisms are called plant microbiota [3] and are formed of bacteria, fungi, and archaea that have adjusted their physiology to environmental demands. This allows the development of microenvironments, and they survive in remote places. Several fungi were studied in detail and are models of some of the routes or molecular mechanisms that activate survival [4].

Alkaline–saline soils are reverted with biofertilizers based on beneficial microorganisms such as *Trichoderma*, which have inhabited environments with extreme conditions for a long time. These strains have genome modifications that give rise to a metabolic adaptation for wellness in their present environment [5]. Where they are located, plants in symbiosis with the filamentous fungi isolated in this work showed increased turgidity; this event is possible due to an effect on moisture retention produced by the hyphae [6].

The physicochemical characteristic conditions of the rhizosphere influence the association, stability, and proliferation of either the plant or microbe and the pH determines the continuity of the mutualistic relationship [7,8]. The pH affects the solubility of macro and micronutrients, which are associated with plants and the growth of their community microbes. At an acidic pH, positive electrostatic charges prevail with the increasing availability of essential elements, such as carbon and nitrogen from organic and mineral matters such as Fe^2+^, Fe^3+^, and Al^3+^, also known as common acid formators [9,10]. This behavior is inverse under alkaline conditions, as well as with chemical compounds with negative electrostatic loads where there is an accumulation of K^+^, Na^+^, Ca^2+^, Mg^2+^, and Cl^−^ in the topsoil [11]. On the other hand, sodification is usually associated with a pH increase in the soil [12]. For fungi, pH is a vital factor that affects growth, development, and competition [13]. These organisms survive environments with extreme pH values. They adapt to their environment with the secretion of extracellular metabolites to mitigate stress; these molecules collaborate with the correct molecular niche formation for the beneficial exchange of protons, which is essential for the modification of valences in chemical compounds. Thus, they are absorbed by microorganisms and plants as macro and micronutrients [14].

The *Trichoderma* genus groups of filamentous fungi are widely known for their biotechnological use as biocontrol agents against several phytopathogenic fungi and their beneficial effect on plant fitness and growth. *Trichoderma* strains are found in soils of various ecosystems growing under multivariate biotic and abiotic conditions. This change is related to their typical fungal morphology and difficult taxonomic identification; this species’ teleomorphic isolates are differentiated in natural environments and in vitro, which presents an entirely different development [15]. In a particular way, for pH variation or extreme values, adapted *Trichoderma* strains present pH control mechanisms with an impact on sensitive changes in the translation of signals, increasing their tolerance to unfavorable conditions and microbiota diversity, positively altering soil properties, and supporting the nutrition of the crop and plant microbial community [16].

Molecular markers become more assertive if a taxonomic description is added with cell scanning electron microscopy (SEM) to describe a possible evolution/speciation/niche selection mechanism that helps discern between species that emerged from a common ancestor. Internal transcribed spacers (ITS) are molecular markers commonly used to distinguish fungal species. However, when sampling sites present adverse conditions that introduce an evolutive factor that promotes microendemism, we proved that it is essential to introduce selected additional primers that reinforce resolutions to distinguish numerous related species and generate a reliable phylogeny [17]. Antibiosis caused by secondary metabolite secretion inhibits the proliferation of various pathogens [18]. The analyzed mechanisms produced in *Trichoderma harzianum* when it is in contact with five fungal pathogens of agroindustrial interest, such as *Fusarium oxysporum, Colletotrichum capsici, Rizoctonia solani, Botrytis cinerea*, and *Gloesercospora sorghi*, revealed an increase in the response of genes previously reported as being involved in mycoparasitism. *Trichoderma* strains are accepted models for invasive microorganism suppression activity and plant protection activation for different interactions, first in the rhizosphere and secondly with associated plants [19]. Worldwide water scarcity elevates the risk of alimentary crisis. Thus, our primary goal is to take advantage of previous experiences in pH response, fungal molecular response, and the interaction of *Trichoderma* species with plants to develop biocompatible strategies to help productive plants adapt to hostile environments for the preservation of alimentary security. 

The study of soils in areas with extreme conditions represents a valuable source of strains with essential variations and adaptations of undescribed molecular strategies that allow them to adapt to current environmental demands. There is sufficient evidence that strains of *T. harzianum* endemic to the arid zone of Mina, Nuevo León, Mexico show adaptations to extreme environments. Therefore, it is vital to study *Trichoderma* from arid soils and its physiological response to alkaline stress to determine how the genus plays a fundamental role as a biological control agent on the phytopathogens of agricultural relevance. Therefore, these strains help to generate healthy microenvironments that allow the establishment of plants due to their potential as biofertilizers; in addition, they positively impact the environment in times of drought caused by climate change.

## 2. Materials and Methods

### 2.1. Soil Collection and Soil Physicochemical Analysis

We carried out a completely random sampling in a 50 m^2^ field with *Agave lechuguilla* and *Fourquieria splendens* (Ocotillo) in Mina, Nuevo León, Mexico (25°59′05.6″/100°37′10.9″). The samples were collected from the main root of healthy plants and the rhizosphere attached to them. Soil extraction was carried out at 0, 15, and 30 cm, and we collected 20 cm roots from each specimen and transported them to 4 °C. The plants were returned to their natural condition. 

The soil of the study area presents uniformity in the delimited area, for which samples were mixed homogeneously, and an external provider determined the physicochemical properties using a method following the statutes of the Official Mexican Standard NOM-021-RECNAT-2002; the specifications included fertility, salinity, and soil classification via studies, sampling, and analysis [20]. The following parameters were evaluated: pH, organic matter, soil texture, and macro and micronutrients.

### 2.2. Fungal Collection 

The Nuevo León *Trichoderma* strains used in this study were isolated using the dilution plate technique. They were incubated in potato dextrose agar (PDA) (Difco Laboratories, Detroit, MI, USA), and *Trichoderma*-selective agar medium (TSM) (1% malt extract, 0.1% yeast extract, 0.02% quitozene, 0.015% Rose Bengal, 0.6 mL of 10% chlortetracycline, 10 mL of 10% chloramphenicol, and 1.5% agar) at 28 °C in darkness. *Trichoderma* strains were identified according to their morphology. All strains were reactivated on PDA, and we recovered conidia for further testing.

### 2.3. DNA Extraction, Molecular Marker Amplification, and Sequencing

Fungal DNA was extracted following the protocol reported by Reader [21]. The polymerase chain reaction was accomplished using the DNAs, and a template was carried out using primers to amplify *ITS* (16S rDNA region), *LSU* (D1–D2 domains of 26/28S), *ACT* (γ-actin), *TUB2* (β-tubulin2), and *TEF1* (section 728–1567) (Appendix A). We evaluated individual and concatenated genes to generate a defined phylogeny, obtaining concrete dendrograms only with the elongation factor *TEF1*. Mytaq DNA polymerase (Meridian Bioscience, Cincinnati, OH, USA) was used to carry out the amplification. The PCR products were purified with the PureLink^®^ Quick Gel Extraction Kit (Invitrogen, Waltham, MA, USA), following the manufacturer´s instructions, and sequenced in the National Laboratory of Genomics for Biodiversity.

### 2.4. Phylogenetic Analysis

DNA sequences were analyzed with BioEdit v. 7.0.9.0, and manual adjustments were made, if necessary, after reviewing the electropherogram. Isolated reference sequences were retrieved from GenBank with homologous features, and then the sequences were aligned with MUSCLE. Phylogenetic analyses of the locus were completed independently using a maximum likelihood model of evolution, with the neighbor-joining method. The statistical support of branches was calculated from 1000 bootstraps.

### 2.5. Tolerance to Alkalinity Stress by Trichoderma Strains

A pH stress sensitivity test was developed by inoculating 1 × 10^6^ spores/mL of *Trichoderma* strains in 0.2× Murashige and Skoog (MS) medium with sucrose as the carbon source. This culture medium is the best candidate because it contains the macro and micronutrients required by plants and microorganisms [22]. When the pH of this medium changes, the availability of the chemical compounds changes. Therefore, we evaluated the absorption of nutrients by the biomass development of *Trichoderma* strains at the initial pH values of 3.0, 6.0, and 9.0. We also evaluated growth at buffer-preserved pH values using a citrate buffer for a pH of 3.0, 2-ethane sulfonic acid (MES) for a pH of 6.0, and N-cyclohexyl-2-aminoethanesulfonic acid (CHES) for a pH of 9.0 (100 mM). Data from three independent experiments were analyzed with an ANOVA to compare all treatments, the incubation was conducted at 28 °C, the quantification of the biomass was carried out for 72 h and the modification was produced by the fungi at the final pH value of the medium.

### 2.6. Scanning Electron Microscopy (SEM) Analysis

SEM scanning was performed as described by [23], with minor modifications. *T. harzianum* and T9_UANL strains (selected in this work) were grown on MS + sucrose in plates with a pH of 6.0 and 9.0 for 72 h at 28 °C. For the SEM analysis, 0.5 cm^2^ of PDA containing each strain at the indicated times was fixed in 3% glutaraldehyde in PBS^®^ (phosphate-buffered saline) for 2 h. Then, plugs were rinsed thrice with cold PBS for 15 min and subsequently post-fixed with 1% OsO^4^ in PBS solution for 1 h, followed by three sequential periods of 15 min in PBS. Finally, samples were mounted and sputter gold-coated in a Cressington model 108 auto and analyzed in a ZEISS model SEM. The SEM was adjusted at 4 kV, 5.5 spots, and a WD of 10 mm. The micrographs were taken with the secondary electron (SE) and circular backscattered detector (CBS) at 1500× magnification.

### 2.7. Mycoparasitism Assays of the Environmental Strains 

For the characterization of the mycoparasites’ antagonism of the strains *T. harzianum* T32 and T9_UANL, the strain *T. atroviride* was added as a model of a good antagonist in this experiment. The phytopathogens *Rhizoctonia solani AG2*, *Rhizoctonia solani AG5*, and *Fusarium oxysporum* were cultivated at 28 °C for 72 h in PDA. Mycelium plugs of *Trichoderma* strains and pathogens were placed approximately 5 cm in front of each other at 28 °C in the dark for ten days.

### 2.8. Trichoderma–Sorghum (Sorghum Bicolor [L]) Interaction and Drought Stress Modulation

*Trichoderma* strains were incubated at 1 × 10^6^ conidia/g in plastic recipients (7 cm diameter × 16 cm height) containing 150 g. We evaluated four conditions (CP: commercial soil (Happy Flower Mexicana, S.A. de C.V, CDMX, Mexico) without *Trichoderma*; CN: arid soil (soil of Mina, Nuevo Leon, Mexico) without *Trichoderma*; AST35: arid soil with *T. harzianum T35*; and ASET9: arid soil with T9_UANL), which were kept in a climatic chamber at 28 °C under a 16 h light (200 μmol m^−2^ s^−1^)/8 h dark photoperiod, and supplied two irrigations of 150 mL non-sterile water per week. 

After five days of strain incubation, we added 25 *Sorghum bicolor* seeds in each lot, which were surface-sterilized in 95% (*v*/*v*) ethanol for five minutes and 20% (*v*/*v*) sodium hypo-chlorite for five minutes and washed five times with sterile water. After 21 d post-interaction, we measured the germination percentage, root and stem length, and fresh weight. 

Stress due to drought and the plasticity that *Trichoderma* strains provide to the *Sorghum* plants were determined according to the procedure described above, which was modified by reducing irrigation entirely for 20 days. On day 21, plants were watered and stored in the growth chamber to observe the recovery of their vigor after 24 h. We measured the germination percentage, root and stem length, and the fresh weight, data from three independent experiments were analyzed with an ANOVA to compare all treatments.

### 2.9. Statistical Analysis 

The IBM SPSS Statistics software statistically analyzed data by one-way analysis of variance. The data represent triplicate determinations from three independent experiments, and the graphs were created in OriginLab 2019.

## 3. Results

### 3.1. Physicochemical Analyses of the Soil

In the study zone of Mina, Nuevo León, Mexico, the main soil texture was sandy clay loam, and the physicochemical characteristics were flawed in providing the optimal nutrition for plant microorganisms (Table 1). Due to the low availability of nutrients and little rainfall, this site is not suitable for crop proliferation. Additionally, the soil had an alkaline pH (8.74 ± 0.24) and was moderately calcareous, with a low organic matter level, which resulted in no carbon availability. Furthermore, based on the parameters of the NOM-021-RECNAT-2002, the compounds that exceed the threshold levels of nutrition were P, Ca, and Na, which at an alkaline pH are toxic to organisms in the rhizosphere.

### 3.2. Isolation and Identification of Trichoderma Strains from the Rhizosphere

We isolated 11 *Trichoderma* isolates from 500 mixed samples between the rhizosphere and root-soil. The highest resolution was obtained with *TEF1*, generating a phylogenetic tree obtained by a neighbor-joining analysis (Figure 1) that demonstrated the following grouping patterns: clade A mainly comprised *T. asperelloides* (T1_UANL, T2_UANL, and T5_UANL), clade B represented two *Trichoderma* sp. isolates (T7_UANL and T10_UANL), and clade C shared ancestry with *T. harzianum* (T3_UANL, T4_UANL, T6_UANL, T8_UANL, T9_UANL, and T11_UANL).

Two strains were isolated from *A. lechuguilla* plants (T4_UANL and T9_UANL), while the largest number of ascomycete strains (T1_UANL, T2_UANL, T3_UANL, T6_UANL, T7_UANL, and T8_UANL) were isolated from the rhizosphere associated with this plant. From *Ocotillo* plants, three strain isolates (T5_UANL, T10_UANL, T11_UANL), with no strains from the rhizosphere of this plant, were found. The results illustrate an excellent variety of *Trichoderma* species that is independent of the sample’s origin. Thus, we reinforced the evidence for the genetic diversity of fungi in arid zones.

### 3.3. Trichoderma Strains Tolerate Alkaline pH

Based on the alkaline rhizosphere and knowing that fungi display a broad tolerance level to different adverse environmental conditions, we evaluated the response of acidic, neutral and alkaline pH in 11 isolated *Trichoderma* strains, using the *T. harzianum* strain as a reference. All strains grew in acidic and neutral pH conditions, whereas only three environmental isolates (*T3_UANL, T6_UANL*, and *T9_UANL*) grew under alkaline conditions (Figure 2). 

In general, when evaluating the growth in the three pH values with the buffer, it was demonstrated that at a pH of 3.0, biomass production after 72 h was evident in all ascomycetes. We affirm that this pH does not affect environmental strains’ mycelial growth. A significant difference was observed when we started at a neutral pH, where the mycelium production by the T9_UANL strain increased to 0.92 g at the end of the experiment, which resulted in the strain with the highest mycelial development at a pH of 6.0. Similarly, the environmental strain that produced greater biomass under alkaline conditions is T9_UANL, showing that the strains that grew in alkaline conditions are tolerant of extreme pH conditions. It is noteworthy that no environmental strain of *T9_UANL* and *T. harzianum T35* modifies the culture medium in the presence of 100 mM buffering conditions. 

It is very important to determine the biomass development and changes in the final pH of the 0.2 × MS culture medium without a buffer. An initial pH of 3.0 in *T. harzianum T35* does not present changes; therefore, it maintains stability in response to the final pH change, while under the same conditions, T9_UANL managed to raise the final pH of the medium to 3.71, a behavior that was not detected in another evaluated strain.

Under a neutral pH condition, T9_UANL has statistically higher biomass production, followed by the control strain *T. harzianum T35*. This behavior is as expected for a pH of 6.0, as demonstrated in the interaction in the media with the buffer. However, the final pH did not present significant changes in the *Trichoderma* strains under neutral conditions.

Data obtained at an initial pH of 9.0 are sparse. Three environmental strains (T3_UANL, T6_UANL, and T9_UANL) generated biomass, which is acceptable because only strains with adaptation or evolution events develop biomass in alkalinity. T9_UANL and T35 belong to the *T. harzianum* species, so we uncovered information that this species is alkaline-tolerant. When evaluating the relationship of biomass production in the strains *T. harzianum* and T9_UANL with the modification of the alkaline pH, we found that both strains modify the pH of the medium to values close to neutrality, a development that is outstanding and different from the rest of the strains evaluated. Therefore, there is enough evidence to use the T9_UANL in later tests, owing to the finding that the influence of alkaline stress reduces the generation of mycelium, and strains adapted to extreme pH conditions are an attractive option for improving environments with a low presence of organic matter due to pH.

### 3.4. Scanning Electron Microscopy

To characterize the morphology of the T9_UANL strain with tolerance and growth under alkaline conditions, we first evaluated its phenotype compared to *T. harzianum T35* at a pH of 6.0. The ideal pH for the development of filamentous fungi is between 5.5 and 6.5; therefore, our control parameter is to observe changes in the hyphae and conidia of the *T. harzianum T35* structure compared to the environmental isolate. We examined well-defined hyphal structures in *T. harzianum T35* (3a), identifying basal septa and phialides for these species of filamentous fungi. In contrast, the strain under study presents turgidity in hyphae, as well as grouping, and atypical remains adhered to the hyphae (Figure 3c). However, it retains a conidiation similar to that reflected in the control strain, which indicates the dispersion of spores under this neutral condition.

We analyzed the morphology by SEM under alkaline conditions as described above. Interestingly, we determined that the hyphae in the control strain presented defined and turgid hyphal structures. We also determined that the fungus groups act as a defense mechanism to mitigate alkalinity, and finally, the *T. harzianum* control strain showed hyperconidiation under these conditions (Figure 3b). Compared to T9_UANL at similar test times and conditions, this particular isolate produced flaccid hyphae; the growth and development phenotype is opposite to that of T35. Therefore, the metabolism reacts differently to mitigate possible hydroxyl ion damage; furthermore, it reflects a sparse topology which indicates it does not cluster to react at an extreme pH (Figure 3d). Additionally, the environmental strain produced a mild conidiogenesis at an alkaline pH attributed to the effect of the microendemism of the characteristics studied with morphologies that are indicative of evolutionary events in *Trichoderma* strains.

The microscopic comparison of both strains confirmed that there is optimal growth in neutral pH values where the sensitivity of the hyphae is not negatively affected; however, under alkaline conditions, there is a deterioration in the development of hyphae, so they present roughness and a decrease in the spaces of union between hyphae. This indicates more significant stimuli due to the alkaline condition that causes phenotypic differences between the *Trichoderma* strains analyzed.

### 3.5. Trichoderma Strains Inhibited Phytopathogen Growth

One of the most studied *Trichoderma* strains is the mycoparasitic activity on various fungi, including many classified as phytopathogens of agroindustrial importance. We confronted T9_UANL against the phytopathogens of the R. solani anastomosis groups AG2 and AG5, and Fusarium oxysporum, using *T. harzianum* and *T. atroviride* as the reference strains. As shown in Figure 4, *T. harzianum* partially limited the growth of all phytopathogens and, to a lesser extent, *R. solani AG2*. T9_UANL presents a pattern of antagonism similar to T35, except that it is less efficient with *R. solani AG2*, the latter extending its mycelium on the colony.

The selected environmental isolate turned out to be an efficient biocontroller; it did not allow the majority colonization of Fusarium oxysporum phytopathogens and presented encapsulating behavior, which was also seen in *T. harzianum* T35 but not observed in *T. atroviride*, and not shown in another confrontation. This behavior indicates that the T9_UANL strain would be effective in mycoparasitic interactions under precarious environmental or agricultural conditions; therefore, *Trichoderma* competes with pathogenic fungi for space to acquire nutrients, and the adaptations generated at the niches of origin promote the generation of metabolites involved in the successful growth of beneficial fungi and prevent the spread of phytopathogens.

### 3.6. Plant–Fungus Interaction and Drought Mitigation

We closely analyzed the effect on the growth of sorghum plants that are of agroindustrial importance with the presence of *Trichoderma* strains directly under endemic environmental conditions.

*Sorghum bicolor* seeds with the ASET9 treatment exhibited a higher germination rate (86%) than the CP treatment (61%). The seeds in interaction with AST35 exhibited a germination rate of 70% (Figure 5); this result is interesting when compared to CN, which showed the lowest germination rate (46%). This reaction defines which strains of *Trichoderma harzianum* promote the germination of *Sorghum bicolor* seeds under this particular condition.

The root length and generation of secondary roots are essential for expanding the surface area for nutrient uptake. The inoculation of *Trichoderma* increased the root system, and the highest growth rates were 48.98% for ASET9 and 29.88% for AST35, as shown in Figure 5. When compared to the CN sample, this result is significant in terms of nutrient uptake efficiency. The presence of environmental strains of *Trichoderma* does not negatively affect plant health. At the same time, it stimulates defense responses and uniformly increases seedling development, which is reflected in the values of the fresh weight in ASET9 and AST35 compared to the sample of CP. 

The amplitude that *Trichoderma* provides to Sorghum plants in arid soil and its mitigation of drought stress is of great interest. The *Sorghum*–*Trichoderma* interaction was subjected to a lack of irrigation for 21 days. The seedlings that were selected regained their vigor after 24 h of irrigation and were not affected by plant architecture. The decrease is apparent at 50% in the height of the plants in all the treatments tested. This effect is directly related to the biomass where a negative variation was observed, a consequence of the lack of nutrients in the soil and drought stress. It was the ASET9 treatment that significantly mitigated the statistically superior drought damage (at 66.56%) compared to CN. The plants presented the most evident damage during this treatment, which demonstrated low values in the variables evaluated. However, AST35 and ASET9 showed acceptable levels of drought mitigation, confirming a 40% higher yield than arid soils and commercial soil.

## 4. Discussion

*Soil conditions* have a direct impact on the isolates of *Trichoderma* strains; Mirkhani [24] found 122 *Trichoderma* strains in forest soils. These types of places present optimal nutrients for the proliferation of organisms, while alkaline soils result in a deficit of nutrients and unfavorable conditions. Zhou [25] found 161 strains of *Trichoderma* with nine different species, among which the most prominent were *T. atroviride, T. pseudoharzianum* and *T. virens*. We isolated 11 strains of *Trichoderma* within 50 m^2^ of Mina, Nuevo León. This information provides indications that the study area has an ecological diversity of microorganisms.

The molecular identification of species of isolates from extreme environments is one of the challenges to overcome because the evolutionary events acquired in the genome of microorganisms alter the grouping by species. Therefore, molecular markers play a vital role in the phylogenetic status of *Trichoderma* and its teleomorphs. Furthermore, internal transcribed spacers (ITS) are no longer suitable for many *Trichoderma* strains [26]. Multi-intron genes that encode nuclear proteins such as *β-tubulin, γ-actin*, and *TEF1* are essential in determining species [27]. We observed that several ribosomal regions could not discern species-level environmental isolates in the genus *Trichoderma*. In addition, we showed that the consensus tree had polytomy with the reference sequences of the molecular markers *TUB2* and *ACT*, in contrast to what was reported by Inglis [28]. 

Furthermore, we agree that *TUB2* and *ACT* are not among the best performing genes. The loci analysis suggests that the simultaneous use of *TEF1*’s large intron and last large exon as diagnostic regions may lead to the most reliable phylogeny [29]. The present study in Mina Nuevo León demonstrated a diversity of species with the molecular marker *TEF1* (Figure 1), whose dominant species were *T. harzianum*, *T. Asperelloides*, and *Trichoderma* sp. The difficulty in discerning between species is due to the nature of the isolates; we assume that the genetic resources available under extreme conditions have distinctive characteristics that do not allow *Trichoderma* species to be grouped together.

The physical–chemical characteristics of the rhizosphere determine the availability of nutrients. These factors are essential for deciding on the survival of plants and microorganisms. The soil studied in the present work has a shortage of macronutrients, micronutrients, and alkaline pH (Table 1). Filamentous fungi grow under acidic and alkaline pH conditions to survive in extreme environments; these fungi adapt their physiology to generate molecules that modify the place where they live. Biomass generation is usually an ability to tolerate alkalinity salinity. *Trichoderma harzianum* T22 Rifai acidifies the medium, allowing solubilization as metallic Mn, Fe, Cu, and Zn oxides and phosphate rock under in vitro conditions [30]. Pelagio [11] reported that when *T. atroviride* acidifies the medium, it causes damage to the germination and development of *A. thaliana* plants when the in vitro interaction was not buffered. Our data indicate that the behavior of *T. harzianum* (T9_UANL) and its biomass growth are not strongly altered under acidic or alkaline conditions. We discovered that this strain has an adaptation to a high pH since, as with other environmental isolates, it has a facility for adapting the pH of the medium to the secretion of organic acids to obtain optimal development (Figure 2). Therefore, the effects of an alkaline pH do not inhibit mycelial growth, and biomass growth is helpful in the establishment of the plant–microbe correlation because mycelium serves to house plant roots and provide nutrients that produce a mutualistic relationship. In this study, growth at an acidic pH is basal for this filamentous fungus, and at an alkaline pH, it is an endogenous adaptation.

When fungi such as *Trichoderma* interact in environments where the pH is unfavorable, they modify their physiology, which causes notable changes in their morphology. This effect is more evident under alkaline conditions than in neutral states. The results show that, in general, the species *T. harzianum* and T9_UANL show stability at a neutral pH, where their cellular processes and the distinctive characteristics of *Trichoderma* are not affected as expected, as shown in Figure 3a,c. The regulation mechanisms in neutral pH values do not modify the structure’s production of hyphae and conidia, which agrees with [31], who mentions that *Trichoderma* strains have the optimal development and formation of conidiophores at pH values close to 6.0. Fungi such as *Trichoderma* assimilate to various growth conditions related to an extreme environmental pH [32]. Physiological alterations in *Trichoderma* strains are due to the differential expression of metabolic signaling genes as the primary strategy to modulate abiotic stress [33]. Our data show that within 72 h of the interaction, the control strain *T. harzianum* presents hyphal turgidity but without modification in the hyphae and phialides at a pH of 9.0. However, in T9_UANL, we note in Figure 3d that 70% of its hyphae are flaccid and have apparent modifications in the structure. Processing extracellular enzymes in fungi represent the biological significance of the morphological changes in alkalinity in environmental strains of *Trichoderma*. Organisms adapted to extreme pH-dependent environments are challenging to classify taxonomically due to changes in their morphology.

The antagonism of the *Trichoderma* genus covers an extensive range of species or is even specific to the pathogen species; this will be given by the molecular and biochemical compatibility of complex metabolisms that produce molecules that generate the biocontrol phenomenon [34]. This work observed that they have different behaviors against specific pathogens for *R. solani AG2*. The strains *T. harzianum* and T9_UANL are better than *T. atroviride* due to their pathogen growth mitigation coverage, which agrees with Nofal [35], who found that *T. harzianum* turns out to be more effective against *Rhizoctonia* compared to *T. atroviride*. On the other hand, both strains of *T. harzianum* (T35 and T9_UANL) exceed the growth of *F. oxysporum* (Figure 4), encapsulating and limiting the development of the phytopathogen. Langa-Lomba [36] reports that *T. harzianum* inhibits the development of this phytopathogen, which has a remarkable effect after six days of interaction. It is crucial to determine the antagonistic capacity of environmental isolates to raise applications against specific pathogens under extreme environmental conditions.

Fungal communities improve the physicochemical properties of the soil, thereby also influencing the nutrient cycle for plant uptake [37]. The *Trichoderma* adapted to extreme environments promotes the generation of fungal biomass that contributes to the strengthening of *Sorghum bicolor* plants, causing a 50% increase in the size of the aerial part and root, as shown in Figure 5a. Soil with saline-alkalinity significantly alters the solubilization of nutrients that are essential for an abundance of microorganisms and plants. Therefore, the addition of *Trichoderma* strains to soils increases the concentration of plant growth-promoting compounds [38]. In addition, the presence of *T. asperellum* in alkaline soils increased the activity of enzymes such as the urease, alkaline phosphatase, sucrase, and peroxidase of hydrogen from maize plants and improved their environment [39]. Drought is abiotic stress that affects plant communities, reduces root development, causes low nutrient acquisition, and directly affects the microbiota present in the rhizosphere [40].

Soil-associated endophytes are essential for adapting plant host spaces and mitigating biotic and abiotic stress [41]. In this work, we develop the plant–*Trichoderma* interaction in arid soils. We confirm what was mentioned by [42], who found that tomato plants treated with *Trichoderma brevicompactum* are 25% more effective in reducing drought. This agrees with what we show in Figure 5b, where the presence of *T. harzianum* reduced the hydric deficit by 50% in *Sorghum* plants. Furthermore, in the data analysis of [43], it was found that the inclusion of *T. harzianum* strains reduces drought in rice plants, and the authors recorded the expression of genes for glutathione metabolism, steroid biosynthesis, carbon metabolism, and photosynthesis pathways, which are critical metabolic pathways for tolerance to this abiotic stress. Therefore, the presence of metabolically adapted fungi in the rhizosphere increases the chemical properties, provides nutrients, generates plant root stability, conserves moisture, prevents damage to plants by drought, and increases the yields of *Sorghum* crops under saline–alkaline conditions, and is a potential biofertilizer for sustainable agricultural projections.

Microbial biofertilizers for the soil represent economic and sustainable alternatives and are feasible to supply the chemical agents used in agriculture [44]. *Trichoderma harzianum* (OL587563) was shown to prevent the mineralization of chemical compounds present in the soil [45]. They are causing changes in the pH, the component most affected by the accumulation of chemical elements in the rhizosphere, causing alkalinization. Likewise, the microorganisms that increase in these extreme conditions present physiological adaptations, which do not modify their growth and production of biomass. In addition, this evolution allows the molecular regulation of genes, as reported by Häkkinen in 2015 [46], where they identify a group of co-regulated genes that are positively regulated at high pH as an ABC transporter, an oxidoreductase, an MFS transporter, a biosynthesis siderophore acetylase, a long-chain fatty acid acyl-CoA ligase, and a non-ribosomal peptide synthase. This leads to having specific microorganisms for each type of soil and agricultural crop.

## 5. Conclusions

In this study, we generated detailed information on the physicochemical properties of the arid soil of Mina, Nuevo León; additionally, in the rhizosphere of two endemic plants, we found the presence of the beneficial fungus of the genus *Trichoderma*, which is subjected to growth. We imitated the alkaline characteristics of where they were obtained, determining that all the strains drastically decrease the initial pH to acidic values that were considered basal in the development of the fungi. We showed that three environmental isolates grow at an extreme pH, described that these physiological changes between species are evolutions acquired under endemic rhizosphere conditions, and showed how pH is one of the factors that most changed the behavior of environmental strains of *Trichoderma*. Finally, we demonstrate that *Trichoderma harzianum* has potential as a biofertilizer for agroindustrial crops in territories with a nutritional deficit and areas with drought problems in Mexico.

## Figures and Tables

**Figure 1 biology-11-01206-f001:**
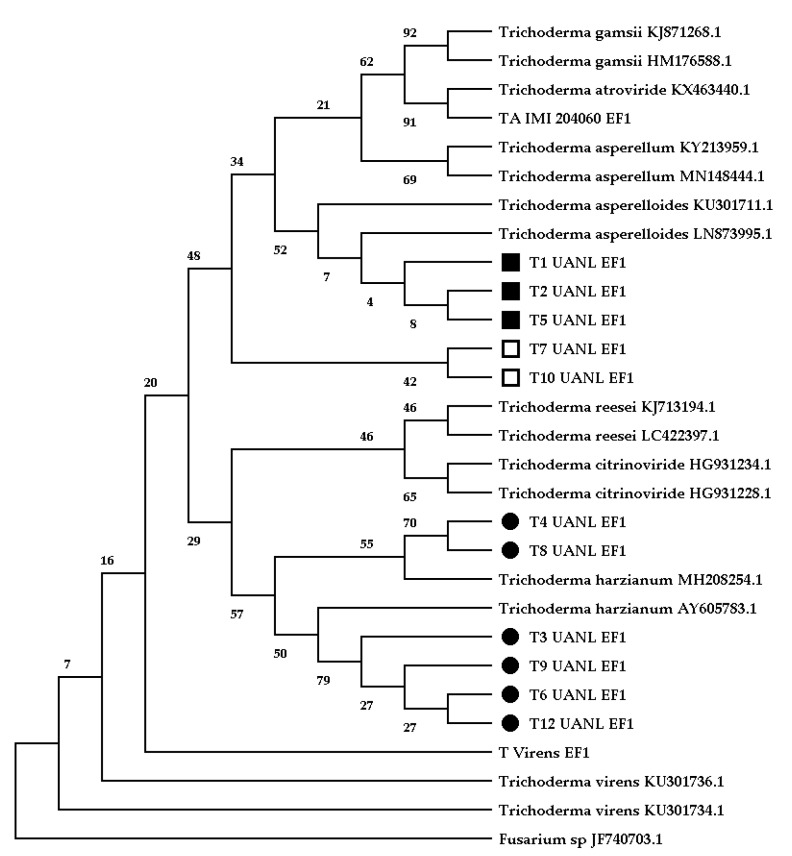
Phylogenetic analysis of *Trichoderma* with the molecular marker TEF1 (section 728–1567) with the neighbor-joining method. The optimal tree with the sum of the branch length = 2.72 is shown. The analysis was performed in MEGA 11. The percentage of replicate trees in which the associated taxa clustered together in the bootstrap test is shown (1000 replicates).

**Figure 2 biology-11-01206-f002:**
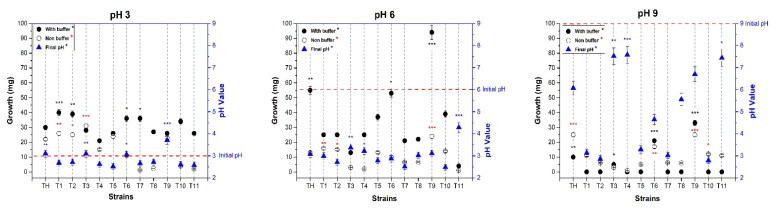
Diversity of the growth of *Trichoderma* in an extreme pH. Biomass production in acidic, neutral and alkaline pH conditions changes the affected strains in 0.2 × MS. The test consisted of three independent experiments and bars indicate standard deviations. The means marked with ***, ** and * are significantly different in each item designated for *p* < 0.05 in Tukey’s test.

**Figure 3 biology-11-01206-f003:**
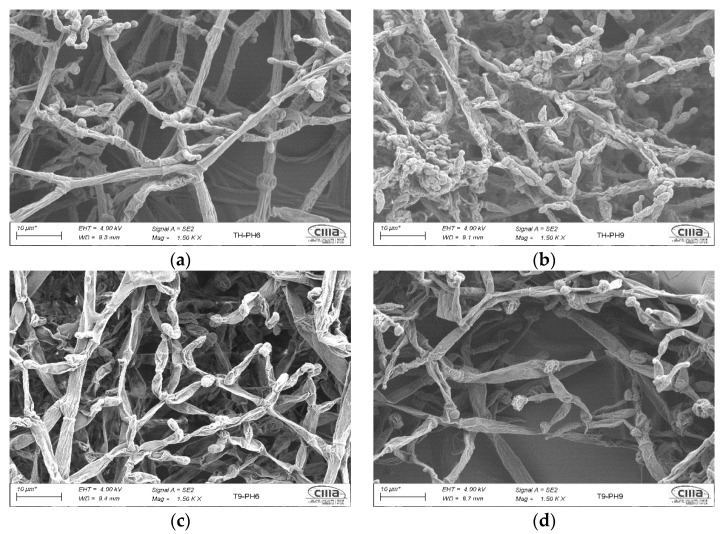
Scanning electron microscopy (SEM). Photomicrographs of (**a**) *T. harzianum* grown in MS + S (control pH of 6.0), (**b**) a stress condition with a pH of 9.0, (**c**) T9_UANL growth in MS + S (control pH of 6.0), and (**d**) a stress condition with a pH of 9.0 at 28 °C for 72 h.

**Figure 4 biology-11-01206-f004:**
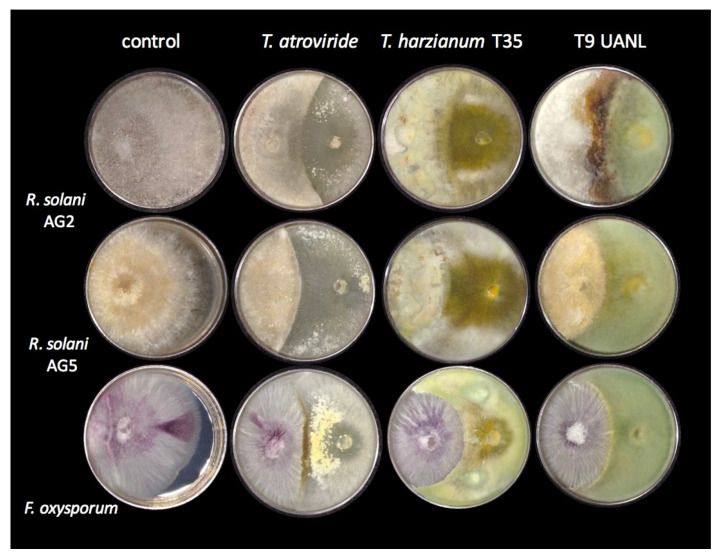
*Trichoderma* strains during mycoparasitic activity. Confrontations of the environmental strain, *T. harzianum T35* and *T. atroviride* against three principal phytopathogens. The test was developed in triplicate in PDA for 72 h at 28 °C.

**Figure 5 biology-11-01206-f005:**
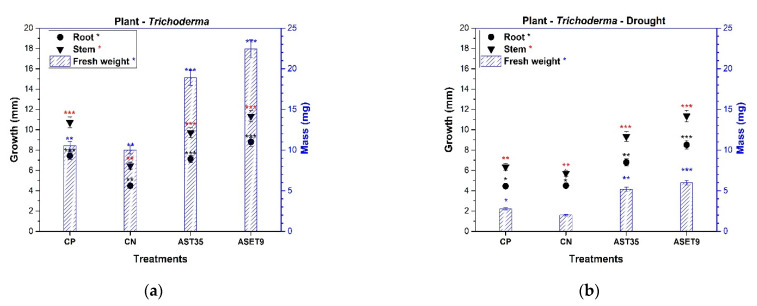
Effect of *Trichoderma* strains in the in vivo plant growth promotion assay of Sorghum bicolor plants in soil with alkaline salinity and drought mitigation. (**a**) Plant–microorganism interaction under endemic agriculture conditions. (**b**) The effect of microorganisms on mitigating stress caused by drought. The means marked with ***, ** and * are significantly different in each item designated for *p* < 0.05 in Tukey’s test.

**Table 1 biology-11-01206-t001:** The abundance of macronutrients and micronutrients in the physical–chemical analysis in the soil of Mina, Nuevo Leon.

**Collected Place**	**Soil Texture**	**pH**	**Organic Matter**	**P-Olsen**	**Ca**	**K**	**Cu**	**CIC**
	(USDA)		(%)	(mg/Kg)	(mg/Kg)	(mg/Kg)	(Mg/Kg)	me/100 g
Mina, NL	Sandy clay loam	8.74 ± 0.24	0.43 ± 0.03	26.1 ± 3.0	2947 ± 9.0	116 ± 24.1	0.22 ± 0.02	15.8 ± 0.01
**Collected Place**	**N-NO3**	**Fe**	**Na**	**Zn**	**Mn**	**Mg**	**B**	**S**
	(mg/Kg)	(mg/Kg)	(mg/Kg)	(mg/Kg)	(mg/Kg)	(mg/Kg)	(mg/Kg)	(mg/Kg)
Mina, NL	10.5 ± 4.69	1.74 ± 0.05	77.5 ± 9.1	0.30 ± 0.06	2.37 ± 1.72	51.2 ± 2.9	0.55 ± 0.11	7.91 ± 2.59

## Data Availability

The total data are described in this manuscript.

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
