# Peer review of "Promotion of Plant Growth in Arid Zones by Selected Trichoderma spp. Strains with Adaptation Plasticity to Alkaline pH"

_biology, 2022, doi:10.3390/biology11081206_

Round 1
Reviewer 1 Report
The manuscript entitled «Promotion of plant growth in arid zone by selected Trichoderma spp. strains with alkaline pH adaptation plasticity» reports about the effect of Trichoderma filamentous fungi on plant growth and development in nutrient deficit soils. The authors have managed to isolate a few Trichoderma sp. strains resistant to alkaline conditions and to analyze their influence on plants.
Authors presented important results in the manuscript, but there are a lot of mistakes and inaccuracies in the text. The authors should thoroughly check and correct a lot of things in the text.
In the first part of “Results” the phylogenetic analysis was performed using TEF1α gene. If authors used fragments of TEF1α gene sequences, it should be clearly described not only in the methods, but in figure legend also. It should be written like TEF1α gene or Tef1α, not like protein. It was mentioned in the text, that the analysis was also performed with other gene sequences. Probably, if the results based on analysis of other genes were similar, it should be mentioned in the text.
Please verify the name of Neighbor-Joining analysis. It should be verified in the text. The same for Latin names of plants in the text.
Please use comma in the end of sentences in the text (not ;).
Line 218. “They were white initially, then turned yellowish and greenish after three days of cultivation”. If it is important for next part of results, the authors should explain it. If it is not, please removing this sentence in the methods.
Line 231. ……in the studied (in the study?).
Line 254. “….considering that the strains that grew in alkalinity are not alkaliphilic, they are considered alkaline tolerant”. This conclusion is not based on your results.
The names of genes should be written in Italics through the text.
Please revise Figure 5 “Rooth” to “Root”. It is not clear from description and the figure legend, which soils the authors used for experiment. It should be clearly defined, which soils were used (peat-most and arid soils without Trichoderma and with Trichoderma or not).
Line 387 and in the last part of text. “The use of genes encoding nuclear multi-intron proteins…”. It should be rephrased, since only genes may be multi-intron.
The Discussion should be more structured.
My suggestion would be to thoroughly check most parts of the manuscript.
Author Response
Dear Prof. Dr.
After carrying out a careful review of our article proposal, based on the suggestions made by the reviewers, we have proceeded to send it for a new evaluation. In the new manuscript, we have highlighted in yellow the modifications made to the original text.
We want to express our sincere thanks to the reviewers for their great work

Reviewer 2 Report
Dear Authors,
The article titled “Promotion of plant growth in arid zone by selected Trichoderma spp. strains with alkaline pH adaptation plasticity” provided a lot of useful information about the effect of endemic Trichoderma species isolated from arid, stressful alkaline soils to promote plant growth and biocontrol on plant pathogenic fungi.
The experiment is planned and implemented. The discussion of the results is correct and clearly. I am suggested to publish this article after making some correction and answering the following comments.
Main issues
Firstly, authors did not explain how they will use or activate the endemic Trichoderma sp especially T9. in the arid soil under investigation.
Moreover, Experimental design about how the Trichoderma sp promote the Sorghum in arid soil is unclear.
Furthermore, Experiment about tolerance of Trichoderma sp to alkalinity is unclear and lacks references
Additionally, there are so many English errors and typing mistake in the entire manuscript that it is make it very difficult to be understood.
Some of these mistakes
The scientific names of species should revise in the whole MS to be italicized.
Line 34; tree or three
Line 108; Change Physic-chemical to Physicochemical (do it in the whole MS)
Line 119; Potato dextrose agar (PDA)
Line 120; Trichoderma selective agar medium (TSM)
Line 121; It is Rose Bengal – also chloramphenicol
Line 142; it is Murashige
Why authors used MS medium which specific for plant growth not fungal growth
Line 151; PDA should write in a full name first at line 119
Line 159; Author didn’t mention from where they get the plant pathogenic fungi (culture collection or they purchased them or isolated them) (Rhizoctonia solani AG2, Rhizoctonia solani AG5, and Fusarium oxysporum).
Line 166; Experimental design about how the Trichoderma sp promote the Sorghum is unclear and doesn’t match with the results. Authors in the results said they use both peat moss and arid soil but they didn’t mention about that in the Materials and Methods.
Author Response

(The authors gave the same response as above.)

Reviewer 3 Report
Regarding the manuscript "Promotion of plant growth in arid zone by selected Trichoderma spp. strains with alkaline pH adaptation plasticity" (ref. biology-1697381) submitted for consideration by the Journal Biology.
In my opinion, while the topic is interesting, the manuscript is not ready for publication. Various problems need to be addressed:
(1) The manuscript has to be proofread by an English native speaker.
A deep English language revision is needed since in its current form, the manuscript is difficult to understand.
(2) The scientific phrasing/wording of this manuscript has to be deeply improved.
For instance, an example can be seen in lines 48 and 49. "...fungi and archaea that have adjusted their physiology to environmental demands and that has allowed to stay..." Allowed to stay? This statement doesn't make sense.
Other example can be seen in line 87 and 88 "Internal transcribed spacer (ITS) are markers of fungal bars" fungal bars???
An additional problematic example can be seen in line 357 "Trichoderma gender". Gender???
(3) There are also multiple other problems throughout the manuscript.
For instance (not an exhaustible list):
-In the abstract it is stated that 500 soil samples were studied. However, this in not stated in the materials and methods section.
-If 500 soil samples were studied why there is only one result for the soil chemical analysis?
-Culture media names are incorrect. For instance, line 119 "dextrose potato agar". Should be Potato dextrose agar (PDA).
-5 genes were amplified and presented as being studied independently and in combination but only information regarding TEF1α is given.
-The phylogenetic analysis is presented as being conducted with Maximum Parsimony, Maximum Likelihood and Bayesian Phylogenetic Inference. However, only a NJ is presented.
-Further details for the procedures conducted are needed.
-All genus and species names should be in italics.
-The work objectives have to be presented in a direct, clear and simple manner.
-The discussion section has to be presented taking into consideration the initial objectives proposed by the authors.
-Some results discussed are absent in the manuscript. For instance lines 353 to 367. The authors discuss the utility of various fungal genetic markers. Yet, these results are never presented in the manuscript.
Author Response

(The authors gave the same response as above.)

Round 2
Reviewer 1 Report
Authors have modified the manuscript, but there are some inaccuracies in the text.
Please rephrase the first sentence in summary (line 12). Authors probably intended to use the phrase such as nutrient deficient soil.
Line 17. Please replace a capital letter in the word “study”.
Please verify the name of Neighbor-Joining method (line 152, line 235).
Please correct the sentence line 387. Authors probably intended to describe the multi-intron genes.
Author Response
Estimado Prof. Dr.
Luego de realizar una cuidadosa revisión de nuestra propuesta de artículo, en base a las sugerencias realizadas por los revisores, hemos procedido a enviarlo para una nueva evaluación.
Presentamos un archivo punto por punto, con los detalles de las revisiones y respuestas del Manuscrito.
Consideramos sumamente valioso el tiempo y la dedicación de los revisores y del editor.

Reviewer 2 Report
Dear authors, a lot of modification has been done make the MS more convenient but some comments you didn't explain it in your revised version. Regards
Author Response
Through this conduit, I send the new corrected version of the work for your consideration.
All the co-authors have carefully read the suggestions, comments, and annotations to the version of the manuscript previously submitted and from which we received a response.
We present a file point by point, with the details of Manuscript revisions and responses.
We consider extremely valuable the time and dedication of reviewers and the editor.

Reviewer 3 Report
Upon reading the new version of the manuscript, it is possible to see that the same English language problems and scientific inconsistencies persist.
For that reason, in my opinion this manuscript remains not suitable for publication.
Author Response

(The authors gave the same response as above.)
